# CMPS: Constrained Mixed Precision Search For Post-Training Quantization

## Abstract

The increasing complexity of deep neural networks (DNNs) requires effective model compression to reduce their computational and memory footprints for deployment on resource-constrained hardware. Mixed-precision search is a prominent bit allocation method based on neural architecture search (NAS) that has been shown to significantly reduce the DNN footprint while preserving the accuracy of the model by allocating bits to each layers based on their quantization sensitivity. However, mixed-precision search is often defined as a dual optimization problem handled with a single heuristic objective function, which does not provide strong guarantees of the resulting compression rate. We propose a post-training reformulation of mixed precision search as an explicit constrained optimization problem, solved using interior-point methods within a framework based on NAS. Our method requires minimal calibration data, as few as 128 samples, in a post-training setting. We corroborate this approach with experiments that span multiple transformer architectures with up to 4 billion parameters, using the MXFP family of data formats. We show that this constrained formulation provides users with higher resolution over compression rates, and we show that explicitly satisfying hardware budgets while optimizing for accuracy can outperform uniform allocation methods, improving performance by up to several standard deviations over the uniform baselines.

## 1 Introduction

The development of efficient Deep Neural Network (DNN) deployment strategies for resource-constrained hardware has led to significant advancements in reduced-precision numerical formats. Researchers and industry consortia have proposed a variety of these formats, including established integer types (e.g., INT8/4), custom floating-point representations (e.g., FP8/6), and block-level schemes such as Block Floating Point (BFP), alongside the more recent Microscaling (MX) family, which encompasses MXFP and MXINT Jacob et al. (2018); Micikevicius et al. (2022); Drumond et al. (2018); Darvish Rouhani et al. (2023); Rouhani et al. (2023). These formats improve arithmetic density and energy efficiency, frequently by amortizing exponent costs across blocks of data. The adoption of low-precision data formats in modern accelerators, such as AMD's Instinct and Google's TPUs, highlights their importance for high-performance inference and training MLCommons (a;b).

Despite progress in format design, a persistent challenge is that DNN exhibits distinct and varied sensitivities to reduced precision and sparsity layer-wise and column-wise (Zhang et al. (2022a); Lee et al. (2024); Dettmers et al. (2022)). This heterogeneity means that uniformly applying a single numerical format across all layers often results in suboptimal trade-offs between model accuracy and computational efficiency. Current post-training quantization approaches for mixed-precision often address this by employing heuristics, such as layer-sensitivity metrics, to guide the allocation of different formats. While very successful and beneficial for their post-training nature—aligning with the established wisdom that training a full-precision model first and then compressing often yields superior results Li et al. (2020)—these heuristic-driven allocations often prefer custom weight encodings over exploiting hardware-provided data formats, leading to compression and decompression steps of weights during inference.

In contrast, Differentiable Neural Architecture Search (DNAS) offers a more principled path, by learning the optimal layer-wise (or block-wise) assignment of hardware-provided low precision data

formats through optimization Wu et al. (2018; 2019); Cai & Vasconcelos (2020; 2021). Historically, DNAS for mixed-precision operated in a training-aware setting. While successfully demonstrating learned allocation, these methods typically handled hardware constraints by incorporating them as soft penalties within a multi-objective loss function. This often requires significant hyperparameter tuning and extensive exploration of the Pareto front to find a desirable balance between accuracy and model size, as the trade-off is not explicitly controlled.

This work seeks to bridge the advantages of both paradigms: the practicality and controllability of post-training methods with the optimized, learned allocation scheme of DNAS that can directly exploit the data formats provided by the deployment device. We revisit the challenge of hardware constraint integration for precision allocation, moving beyond soft penalties. We reformulate the problem within a DNAS-inspired post-training framework, treating hardware requirements as explicit, hard constraints. Our approach directly integrates model complexity into the optimization objective using a barrier-based interior-point method Boyd & Vandenberghen (2004); Nocedal & Wright (2006), systematically driving solutions towards feasibility while enabling fine-grained control over the final model complexity. This allows for a learnable, yet rigorously constrained, bit allocation in a post-training setting.

Our framework targets the post-training setting, using frozen weights and only a small calibration dataset. It operates on relaxed softmax-distributed architectural parameters and employs an annealed regularization schedule to efficiently solve the constrained optimization problem. This yields format allocations closely adhering to resource budgets while optimizing performance. Notably, our approach offers finer resolution in compression ratios and predictable model behavior across resource settings without retraining or finetuning. Our contributions are as follows:

- We introduce a novel, principled interior-point optimization framework for post-training, hardware-constrained bit allocation. This DNAS-inspired approach requires no model retraining or fine-tuning and operates effectively with as few as 128 calibration samples.

- Our framework provides fine-grained control over model compression, enabling stable and predictable performance across a higher resolution of intermediate compression ratios than typically available with fixed hardware format choices, bridging performance gaps.

- We demonstrate that our method enables smart, layer-wise precision allocation that goes beyond uniform quantization. This results in significant performance gains; for instance, with an average of only 4.5 effective bits, we achieve perplexity reductions of up to 9.06 points on C4 (Qwen2.5-1.5B) in few-shot settings and accuracy improvements of up to +10.7% (Qwen2.5-0.5B) in zero-shot evaluations compared to 4.25-bit uniform MXFP4 on the OpenAI Lambada benchmark.

- Crucially, our approach even allows models to outperform higher-precision uniform baselines using fewer bits. For example, our 4.5-bit mixed-precision Gemma models surpass both 6.25-bit and 8.25-bit uniform MXFP allocations on C4 perplexity and WikiText perplexity, showcasing substantial memory gains without sacrificing, and sometimes even improving, performance.

## 2 RELATED WORK

**Low precision data formats**  The imperative to reduce the computational and memory footprint of Deep Neural Networks (DNNs) has spurred significant advancements in low-precision data representations. This area has seen a progression from conventional fixed-point integers (e.g., INT8, INT4) and custom floating-point types (e.g., FP8, FP6) towards more sophisticated block-based numerical formats (Jacob et al. (2018); Micikevicius et al. (2022); Gholami et al. (2021)). Block-based formats, which group multiple elements under a shared exponent or scaling factor, are particularly prominent as they improve arithmetic density while preserving essential dynamic range. For example, Drumond et al. (2018) proposed Hybrid Block Floating Point (HBFP), a representation where dot product operations utilize block floating point arithmetic, while element-wise functions and control logic retain the standard floating point. They showed that this hybrid strategy ensures comparable convergence to full-precision training (FP32) across various workloads and can achieve significant throughput gains with modest hardware modifications, positioning HBFP as a viable drop-in replacement.

More recently, Darvish Rouhani et al. (2023); Rouhani et al. (2023) developed the Microscaling (MX) data format family to support both inference and training. The MX formats combine narrow data types (e.g., INT8, FP6, FP4) with fine-grained block-level scaling. The authors designed these formats for high-performance computing environments, providing a tunable balance between computational efficiency, numerical stability, and usability. Their evaluations demonstrate that MX formats maintain model fidelity even for large-scale transformers using 8-bit and lower representations for activations, weights, and gradients. Furthermore, they show compatibility with common training pipelines and minimal need for hyperparameter or infrastructure adjustments. Consequently, MX formats are receiving considerable attention and support in next-generation hardware architectures (Open Compute Project (2023); Samson et al. (2024)).

**Differentiable Neural Architecture Search**   The optimal bit allocation problem for neural networks is the following: Let a generic neural network of $L$ layers labeled $l \in \mathbb{L} = \{1, ..., L\}$ with associated weight tensors $W = \{W^l\}_{l=1}^L$. Assuming we have access to a set of quantization functions labelled $Q_d(.)$ with $d \in \mathbb{D} = \{1, ..., D\}$, we define:

$$\hat{W}^l \overset{def}{=} \sum_d^D A_d^l Q_d(W^l)$$

$$s.t. \quad \sum_{d=1}^D A_d^l = 1 \quad \forall l \in \mathbb{L} \tag{1}$$

$$A_d^l \in \{0, 1\} \quad \forall (l, d) \in \mathbb{L} \times \mathbb{D}$$

Equation (1) represents the core formulation used in differentiable mixed-precision search frameworks. Wu et al. (2018); Cai & Vasconcelos (2020; 2021); Wu et al. (2019); Clark et al. (2018) aim to find the optimal decision variables $A_d^l$ solving the multi-objective minimization problem:

$$\min_{\hat{W}, A} \quad \mathcal{L}(A, \hat{W})$$

$$\min_A \quad \mathcal{C}(A) \tag{2}$$

Here, $\mathcal{L}(A, \hat{W})$ is the model loss function, and $\mathcal{C}(A)$ is defined as a complexity cost on the architecture $A$, often related to hardware constraints such as size or latency. Throughout this paper, we set $\mathcal{C}(A)$ as the average bit width per element of the target model. The exponential number of possibilities for a choice of $A$ and the latency induced by the evaluation of $\mathcal{L}$ make it difficult to efficiently solve the problem using combinatorial techniques. Taking inspiration from approximation algorithms, a popular approach is to relax the conditions on $A$ and reinterpet $A^l$ as a probability distribution.

$$\begin{cases} \sum_{d=1}^D A_d^l = 1 \\ A_d^l \in \{0, 1\} \end{cases} \implies \begin{cases} \sum_{d=1}^D A_d^l = 1 & \forall l \in \mathbb{L} \\ A_d^l \geq 0 & \forall (l, d) \in \mathbb{L} \times \mathbb{D} \end{cases} \tag{3}$$

To that end, Neural Architecture Search frameworks introduced the following parameterization of $A^l$ in terms of logits $\{x^l\}_{l=1}^L \subset \mathbb{R}^D$:

$$A_d^l \equiv A_d^l(x^l) = \frac{exp(x_d^l)}{\sum_{k=1}^D exp(x_k^l)} \tag{4}$$

NAS-inspired frameworks for mixed precision quantization leverage the above relaxation by building a super-network, for which they train the weights and architectural parameters alternatively, handling the search for two sets of parameters at once. Once done, a feasible solution $\bar{A}$ to the original decision problem is rounded from the learned solution $A$ to the relaxation by sampling each layer's bit-width from the learned distribution $A^l$. Cai & Vasconcelos (2021) propose to instead

select, for each layer, the quantization option with maximum probability, where the data format with the largest associated parameter is sampled. Wu et al. (2019); Wan et al. (2020) also propose the use of the Gumbel-Softmax to simulate random categorical sampling steps during the search phase and enforce the convergence of the distribution by scheduling the "temperature" hyperparameter of the Gumbel-Softmax function. Most importantly, Yu et al. (2020) first introduced the intuition of incorporating hardware constraints via an additive barrier penalty within such relaxed search frameworks, our work revisits these foundational assumptions by deriving the optimization objective directly from Lagrangian principles and perturbed Karush-Kuhn-Tucker conditions to develop a novel post-training algorithm.

## 3 CONSTRAINED MIXED PRECISION SEARCH

**Post-training compression**   While neural architecture search for mixed-precision models has mostly been developed as part of a quantization-aware training framework, Li et al. (2020) have shown that the optimal approach to model compression is to first train large networks in full precision, and then aggressively compress the model for deployment. By following this framework, we rework on the assumptions of the neural network bit-allocation problem. We propose to restrict the problem to the search of architectural parameters for pre-trained models, maintaining the previously learned weights $W^*$ frozen. This significantly reduces the number of parameter updates, as the number of architecture parameters only grows linearly with the depth of the associated network and is invariant with respect to the dimensions of its layers. In turn, this allows us the use of a significantly smaller calibration dataset for the search phase.

**User-defined architectural constraints**   In practical deployment scenarios, models must conform to diverse hardware constraints, including limits on total model size, supported numerical formats, and compute budgets such as FLOPs or BOPs. These constraints are platform-dependent and are often non-negotiable. To accommodate such deployment requirements, we reformulate the bit allocation task not as a multi-objective trade-off between accuracy and complexity, but as a constrained optimization problem. Let $\mathcal{C}(.)$ be a differentiable architectural cost function e.g., total model size in bits as follows:

$$\mathcal{C}(A) = \frac{\sum_{l=1}^{L} s_l \sum_{d=1}^{D} A_{l,d} b_d}{\sum_{l=1}^{L} s_l} \tag{5}$$

where $L$ is the number of layers, $D$ is the number of format choices, $s_l$ is the number of parameters in layer $l$, $b_d$ is the bit-width of format choice $d$, and $A_{l,d}$ is the continuous, relaxed weight for choosing format $d$ for layer $l$.

And let $B$ denote an upper bound imposed by the hardware (e.g., a target average bit-width). Our goal is to minimize the loss subject to this constraint:

$$\begin{aligned} \underset{A}{minimize} \quad & \mathcal{L}(A, \hat{W}^*) \\ subject\ to \quad & \mathcal{R}_B[A] = B - \mathcal{C}(A) \geq 0 \end{aligned} \tag{6}$$

This constrained formulation enables the principled integration of hardware-awareness into the precision allocation process, ensuring that the resulting architecture is both accurate and deployable.

**Interior-Point Formulation**   To solve the non-linear constrained optimization problem defined in equation 6, we employ techniques common in constrained optimization, specifically adopting a barrier-based interior-point method. The first step involves formulating the Lagrangian function, which incorporates the objective function and the constraint scaled by a Lagrange multiplier $\lambda$:

$$\mathcal{L}_\lambda(A) = \mathcal{L}(A, \hat{W}^*) + \lambda \mathcal{R}_B[A] \tag{7}$$

Here, $\mathcal{L}(A, W^*)$ represents the original loss function (our objective to minimize) with fixed model weights $W^*$, and $\mathcal{R}_B[A]$ represents the hardware constraint function (which must be non-negative, $\mathcal{R}_B[A] \geq 0$). The variable $\lambda$ is the Lagrange multiplier associated with this inequality constraint.

For a given candidate solution $A^*$ to be a local optimum of the constrained problem equation 6 (under certain regularity conditions), it must satisfy the Karush-Kuhn-Tucker (KKT) conditions. These conditions are fundamental necessary conditions for optimality in nonlinear programming Boyd & Vandenberghen (2004). They generalize the method of Lagrange multipliers to handle inequality constraints. For our problem, the KKT conditions are:

$$
\mathbf{KKT} \begin{cases}
\nabla \mathcal{L}(A^*, \hat{W}^*) + \lambda \nabla_A \mathcal{R}_B[A^*] = 0 & \text{(Stationarity)} \\
\mathcal{R}_B[A^*] \geq 0 & \text{(Primal Feasibility)} \\
\lambda \geq 0 & \text{(Dual Feasibility)} \\
\lambda \mathcal{R}_B[A^*] = 0 & \text{(Complementary Slackness)}
\end{cases}
\tag{8}
$$

However, in practice, satisfying the strict complementarity condition $\lambda \mathcal{R}_B[A^*] = 0$ leads to optimization challenges due to discontinuity at the boundary of the feasible region. To circumvent this, we adopt the perturbed KKT formulation, commonly used in interior point methods, which replaces the complementarity condition with a small nonzero slack $\mu$.

$$
\mathbf{KKT}(\mu) \begin{cases}
\nabla \mathcal{L}(A^*, \hat{W}^*) + \lambda \nabla \mathcal{R}_B[A^*] = 0 \\
\mathcal{R}_B[A^*] \geq 0 \\
\lambda \geq 0 \\
\lambda \mathcal{R}_B[A^*] = -\mu
\end{cases}
\qquad s.t. \quad \mu \to 0
\tag{9}
$$

This relaxation smooths the boundary behavior of the optimizer and permits convergence to the constrained optimum from within the feasible set. Following this approximation, the Lagrange multiplier $\lambda$ can be reformulated as:

$$
\lambda = \frac{-\mu}{\mathcal{R}_B[A^*]}
\tag{10}
$$

Plugging $\lambda$ into the gradient of equation 7 induces the gradient of a logarithmic function:

$$
\nabla \mathcal{L}_\lambda(A, \hat{W}^*) = \nabla \mathcal{L}(A, \hat{W}^*) - \mu \nabla \mathcal{R}_B[A] \frac{1}{\mathcal{R}_B[A]}
\tag{11}
$$

It corresponds to minimizing the following barrier-augmented objective:

$$
\underset{A}{\text{minimize}} \quad \mathcal{L}(A, W^*) - \mu \ln(\mathcal{R}_B[A])
\tag{12}
$$

Here, the logarithmic barrier $\ln(\mathcal{R}_B[A])$ diverges to $-\infty$ as the constraint approaches 0, effectively discouraging the optimizer from leaving the feasible region. In the appendix, we also propose a surrogate constraint function $\hat{\mathcal{R}}$ to smooth out the search near the boundary of the feasible region.

**Algorithm** Following our previous derivation, we propose the following iterative algorithm.

To minimize the model loss while staying within region defined by the memory constraint, the interior-point method adds a logarithmic barrier penalty. This penalty acts like a repulsive force that becomes very large at the constraint boundary, ensuring the solution always remains strictly feasible prior to the rounding step, which may induce negligible variations. Optimization starts with a strong repulsion (large $\mu$), keeping the solution near the center of the feasible region. As $\mu$ is gradually decreased across iterations, the barrier's influence weakens, allowing the solution to follow a "central path" closer to the true minimum of L while still being repelled from the boundary. The figure below illustrates how each step balances minimizing the objective (descent step) with staying feasible (step towards central path), ultimately converging to the constrained optimum as $\mu \to 0$.

---

**Algorithm 1** Constrained Bit Allocation

---

**Require:** Pretrained weights $W^*$, initial (relaxed) allocation parameters $A^{(0)}$, calibration dataset $\mathcal{D}_{\text{cal}}$, loss function $\mathcal{L}(\cdot, \cdot; \mathcal{D}_{\text{cal}})$, softplus constraint surrogate $\mathcal{R}_B[\cdot]$, initial barrier weight $\mu > 0$, decay factor $0 < \delta < 1$, number of outer iterations $T$, number of inner epochs $E$, learning rate for $A$ parameters $\eta_A$.

**Ensure:** Final discrete allocation $\hat{A}$ satisfying $C(\hat{A}) \leq B$

1: $t \leftarrow 1$
2: $\mu^{(1)} \leftarrow \mu$
3: $A \leftarrow A^{(0)}$
4: **while** $t \leq T$ **do**
5:   **for** $epoch = 1$ to $E$ **do**
6:     $g_A \leftarrow \nabla_A \left[ \mathcal{L}(A, \hat{W}^*; \mathcal{D}_{\text{cal}}) - \mu^{(t)} \ln \left( B - C(A) \right) \right]$
7:     $A \leftarrow A - \eta_A g_A$
8:   **end for**
9:   $\mu^{(t+1)} \leftarrow \delta \mu^{(t)}$
10:   $t \leftarrow t + 1$
11: **end while**
12: $\hat{A} \leftarrow \text{round}(A)$
13: **return** $\hat{A}$

---

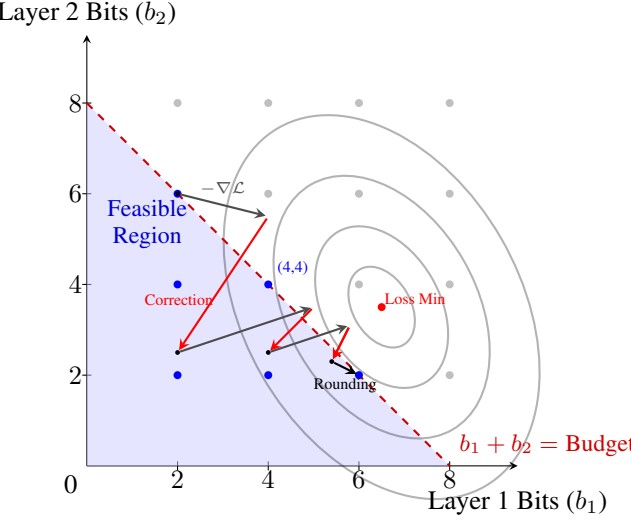

Figure 1: Search space for bit allocation in a 2-layer network. Axes are bits per layer $(b_1, b_2)$. Blue dots are feasible discrete choices under a budget constraint (shaded region, $b_1 + b_2 \leq$ Budget). Gray dots are infeasible. Gray ellipses represent level curves of the loss function, with the true (continuous) minimum marked in red, located between discrete points.

## 4 EXPERIMENTAL METHODOLOGY AND RESULTS

To evaluate the effectiveness of our method, we conducted a series of post-training experiments on 10 transformer models of varying scales, including OPT (Zhang et al. (2022b)), LLaMA (Grattafiori et al. (2024)), Gemma (Team et al. (2024; 2025)), and Qwen (Team (2024a;b); Qwen et al. (2025)) architectures with up to 3 billion parameters. All experiments follow the setup for causal language modeling provided by the open-source examples of HuggingFace. Importantly, we operate strictly in the post-training regime: the model weights are frozen, and only the architectural parameters governing format assignment are optimized using a small calibration set. All models are adapted using a mixture of MX-compliant formats, specifically MXFP4 and MXFP8. All experiments were performed on a single GPU and mainly implemented using the PyTorch library, Huggingface's transformers library, and MXFP emulation library by Microsoft (2024).

**Empirical study 1: few-shot scenario** In the few-shot setting, we use a small calibration set of 128 samples drawn from the training split of the C4 corpus proposed by Dodge et al. (2021). We then evaluate perplexity (PPL) on the C4 validation split, gauging performance on in-distribution data under low-data conditions. Figure 2 compares the perplexity of our mixed-precision allocations constrained to an average of 4.5 bits ('Mixed', red cross) against uniform MXFP baselines. Remarkably, this technique, initially designed to amortize the performance degradation of low-precision quantization, demonstrates that a carefully constrained 4.5-bit allocation can be competitive with, and in some cases, even outperform higher-precision uniform allocations. For instance, Gemma-3-4B-it with our 4.5-bit mix surpasses both uniform MXFP6 and MXFP8 baselines on C4 perplexity. Significant perplexity improvements are also observed for Llama-3.2-1B (↓2.60) and Gemma-3B-it (↓3.67), with both models effectively closing the performance gap to their uniform MXFP6 counterparts despite our mixture being predominantly composed of MXFP4 tensors. Other notable improvements of over 5% perplexity reduction include OPT-350M (↓1.5 points) and Qwen2.5-0.5B (↓2.08 points) when compared to the MXFP4 baseline. These gains are achieved with only an approximate +0.25 effective bit increase.

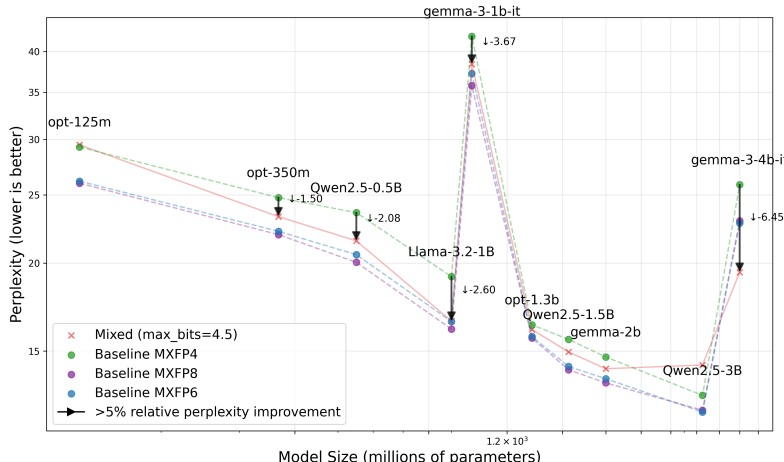

Figure 2: Few-shot perplexity (lower is better) on C4 validation set for models constrained to 4.5 bits. Our mixed allocation (red cross) is compared against uniform MXFP4, MXFP6, and MXFP8 baselines. Black arrows indicate perplexity improvements $\geq 5\%$ improvements over the MXFP4 baseline.

Figure 3 shows the achieved effective bit-widths for the models under the 4.5-bit constraint. In 8 out of 10 models, the resulting effective bit-widths are strictly under the user-defined constraint. For OPT-125M and Qwen2.5-0.5B, the rounding step in the final allocation resulted in a slight increase, but the framework generally adheres closely to the target hardware budget, with most deviations being minor.

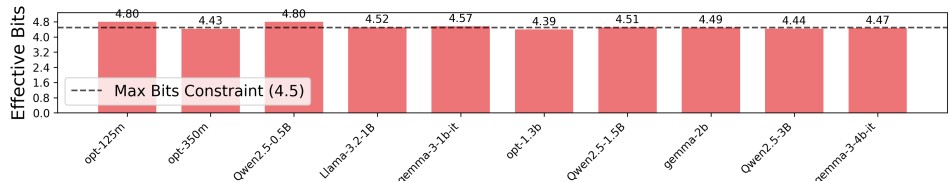

Figure 3: Effective bits achieved for models constrained to 4.5 bits in the few-shot setting. The dashed line indicates the target constraint.

**Empirical study 2: zero-shot scenario** For zero-shot evaluation, we calibrate using 256 samples from the C4 training split. We then employ the lm-eval-harness benchmark suite Biderman et al. (2024); Gao et al. (2024) to assess model performance on downstream tasks without task-specific

fine-tuning. We report accuracy on LAMBADA Paperno et al. (2016) (predicting the last word of a passage, testing context understanding) and perplexity on WikiText-2-raw (general language modeling capability) (Merity et al. (2016)).

Figure 4 presents LAMBADA accuracy for the 4.5-bit constraint. Our mixed allocation yields substantial accuracy improvements over the uniform MXFP4 baseline, including a striking +10.7% for Qwen2.5-0.5B. Other significant gains (exceeding two standard deviations) are seen for OPT-350M (+2.7%) and Gemma-3-1B-it (+3.8%), with the latter closing the gap to the MXFP6 baseline. Impressively, for Qwen2.5-1.5B, Gemma-2B, OPT-1.3B, and Qwen2.5-3B, our 4.5-bit mixed-precision models outperform both the uniform MXFP6 (6.25 bits) and MXFP8 (8.25 bits) baselines, showcasing the power of strategic bit allocation.

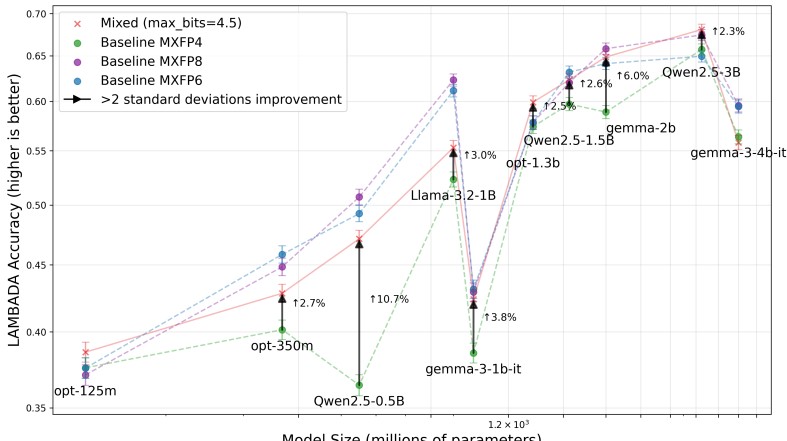

Figure 4: Zero-shot LAMBADA accuracy (higher is better) for models constrained to 4.5 bits. Our mixed allocation (red cross) compared against uniform baselines. Black arrows indicate accuracy improvement over MXFP4 baseline exceeding 2 standard deviations.

We further evaluate on WikiText perplexity (results shown in Figure 5). Consistent with other findings, we observe more than 5% perplexity improvements across the board. Notably, the Gemma family excels: Gemma-2B at 4.5 bits outperforms the MXFP4 baseline by 2.9 perplexity points and the MXFP6 baseline by over 45 points. Similar strong outperformance of the 6.25-bit MXFP6 baseline is seen with Gemma-3-1B-it (↓5.80 vs MXFP4) and Gemma-3-4B-it using only 4.5 effective bits. For other models, such as OPT-125M (↓3.31), OPT-350M (↓2.14), and Qwen2.5 (↓2.46), our method successfully closes the performance gap to higher-precision uniform allocations. These accuracy and perplexity improvements, achieved with frozen weights and calibration solely on C4 data, underscore the generalization capability of the learned allocations.

These zero-shot findings underscore that targeted precision allocation, guided by our constrained optimization, effectively preserves crucial model capabilities for downstream tasks. This approach often yields significantly better results than uniform low-bit formats, particularly under tighter memory constraints, and can even rival or exceed the performance of higher-precision uniform schemes. The method provides practitioners fine-grained control over the accuracy-compression trade-off simply by selecting the appropriate average bit-width constraint.

## 5    DISCUSSIONS

**Strengths**    A key strength of this methodology lies in its combined practicality, efficiency, and theoretical robustness. The entire allocation process operates post-training, requiring no modification or fine-tuning of the original model weights, which significantly reduces computational cost and technical complexity. Furthermore, its reliance on a remarkably small number of calibration samples (e.g., 128-256) and minimal runtime (down to 2.5 minutes for OPT-125M and up to 32 minutes for Gemma-3-4B) makes it highly feasible even with limited data access. Despite this efficiency, the derived allocations exhibit robust generalization, improving performance on both in-distribution

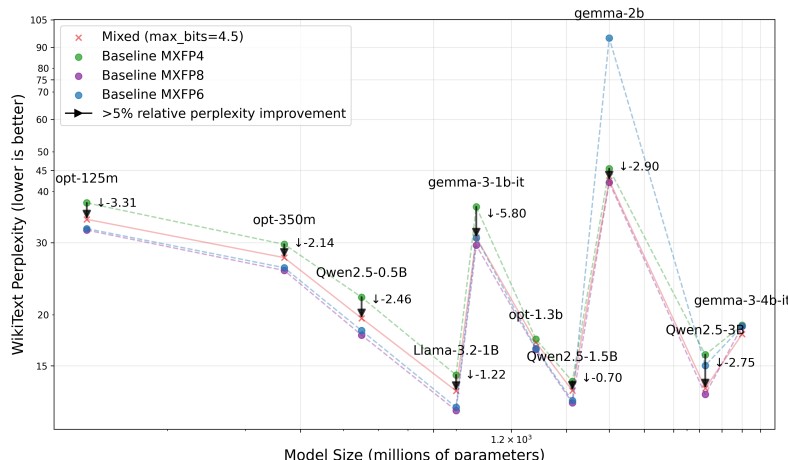

Figure 5: Zero-shot wikitext perplexity (lower is better) for models constrained to 4.5 bits. Our mixed allocation (red cross) compared against uniform baselines.

and diverse zero-shot tasks. From a theoretical standpoint, formulating the problem as a constrained optimization provides a more rigorous foundation than heuristic or multi-objective techniques lacking strong guarantees or requiring complex Pareto front analysis. The well-established interior-point method, applied with an average bit-width constraint, offers an interpretable way to directly incorporate hardware budgets.

**Limitations**   We acknowledge certain limitations. The current approach implicitly utilizes a "supernet" concept during the search phase, where gradients for different format assignments are needed. This can temporarily increase memory usage during the allocation optimization compared to standard inference. However, the search often involves low-bit formats; the memory required to hold activations or gradients for multiple low-bit options might still be comparable to, or less than, holding a single higher-precision (e.g., FP16 or BF16) baseline tensor.

**Future work**   Optimizing the memory of the supernet and the computational efficiency of the allocation search itself presents a viable avenue for future research. Investigating the interplay between different types of constraints (e.g., latency-aware constraints) and granularity (e.g., column-wise rather than layer-wise) within this framework is another promising direction.

**Societal Impact**   By enabling more efficient deployment of large neural networks, this work contributes to reducing the energy consumption and computational resources required for AI model inference. This can lead to more sustainable AI practices and broader accessibility to advanced AI capabilities on less powerful hardware, potentially mitigating environmental impact and democratizing access to technology.

## 6   CONCLUSION

The escalating complexity of deep neural networks demands efficient deployment on resource-constrained hardware. While mixed-precision formats offer substantial potential, optimal layer-wise allocation under hardware constraints has remained a key challenge, often addressed by heuristics or training-aware searches. This paper introduced a principled, post-training framework for mixed-precision allocation grounded in constrained optimization theory. Our interior-point method directly incorporates hardware limitations, like memory footprint via average bit-width constraints, using only small calibration datasets and without requiring model retraining. Empirical evaluations across diverse transformer architectures demonstrate that strategic allocation, even with a limited set of precision formats (e.g., MXFP4 and MXFP8), bridges performance gaps between intermediate-precision formats (e.g., MXFP6), and can even outperform high-precision configurations.

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

## A   Heuristic Surrogate Function for Smoother Search

The optimization of architectural parameters $A$ using stochastic gradient descent based optimizers for our constrained bit allocation problem introduces a practical challenge. The logarithmic barrier term, $-\mu \ln(\mathcal{R}_B[A])$, where $\mathcal{R}_B[A] = B - C(A, W^*)$ is the constraint function, is undefined or tends to $-\infty$ if the constraint is violated (i.e., if $C(A, W^*) \geq B$, making $\mathcal{R}_B[A] \leq 0$). With SGD, due to the inherent noise in gradient estimates or potentially poor initialization of $A$, updates might inadvertently lead to iterates $A^{(t)}$ that approach or even violate this constraint boundary. Such violations can cause numerical instability (e.g., 'log(0)' or 'log(negative number)') and large, uninformative gradients, hindering the convergence of the optimization process.

To address this, we propose a surrogate function $\hat{\mathcal{R}}_B[A]$ for the constraint $\mathcal{R}_B[A]$ that behaves more gracefully near and beyond the constraint boundary. Ideally, this surrogate should satisfy the following properties:

1. When the constraint $C(A, W^*) < B$ is satisfied (i.e., $\mathcal{R}_B[A] > 0$), the surrogate $\hat{\mathcal{R}}_B[A]$ should approximate the original constraint function $\mathcal{R}_B[A]$.

2. When the constraint $C(A, W^*) \geq B$ is violated (i.e., $\mathcal{R}_B[A] \leq 0$), the surrogate $\hat{\mathcal{R}}_B[A]$ should smoothly approach 0 from the positive side ($0^+$). This ensures that $\ln(\hat{\mathcal{R}}_B[A])$ remains defined and tends to $-\infty$, preserving the barrier effect without encountering numerical errors associated with non-positive arguments.

3. The surrogate function should be differentiable to allow for gradient-based optimization.

A suitable candidate that fulfills these requirements is a scaled and shifted softplus function. The softplus function is a smooth approximation of the rectifier function $\text{ReLU}(x) = \max(0, x)$. We define our surrogate constraint function $\hat{\mathcal{R}}_B[A]$ as:

$$\hat{\mathcal{R}}_B[A] = \beta \cdot \text{softplus}\left(\frac{\mathcal{R}_B[A]}{\beta}\right) = \beta \cdot \ln\left(1 + \exp\left(\frac{B - C(A, W^*)}{\beta}\right)\right) \tag{13}$$

where $\beta > 0$ is a temperature parameter that controls the smoothness of the approximation. It is worth noting that the softplus function is readily available in popular deep learning libraries such as PyTorch (as 'torch.nn.functional.softplus'). In our implementation, we can set $\beta = \mu^{(t)}$, the current barrier weight, allowing the sharpness of the surrogate to anneal along with the barrier itself. For simplicity in the main text, we denoted this as $\mu \ln(1 + \exp((B - C(A, W^*))/\mu))$, which corresponds to Equation 13 with $\beta = \mu$.

Let's verify the asymptotic properties of this surrogate $\hat{\mathcal{R}}_B[A]$ as defined in Equation 13, particularly focusing on its behavior as the original constraint $\mathcal{R}_B[A]$ varies:

**Case 1: Constraint is well satisfied ($\mathcal{R}_B[A] \gg 0$)**   When $B - C(A, W^*) \gg 0$, then $\frac{\mathcal{R}_B[A]}{\beta} \gg 0$. In this regime, $\exp\left(\frac{\mathcal{R}_B[A]}{\beta}\right)$ is very large. So, $\ln\left(1 + \exp\left(\frac{\mathcal{R}_B[A]}{\beta}\right)\right) \approx \ln\left(\exp\left(\frac{\mathcal{R}_B[A]}{\beta}\right)\right) = \frac{\mathcal{R}_B[A]}{\beta}$. Therefore,

$$\hat{\mathcal{R}}_B[A] = \beta \cdot \text{softplus}\left(\frac{\mathcal{R}_B[A]}{\beta}\right) \approx \beta \cdot \frac{\mathcal{R}_B[A]}{\beta} = \mathcal{R}_B[A] = B - C(A, W^*). \tag{14}$$

This shows that when the constraint is comfortably met, the surrogate closely approximates the original constraint function.

**Case 2: Constraint is violated or at the boundary ($\mathcal{R}_B[A] \ll 0$ or $\mathcal{R}_B[A] \approx 0$)**   When $B - C(A, W^*) \ll 0$ (constraint significantly violated), then $\frac{\mathcal{R}_B[A]}{\beta} \ll 0$. In this regime, $\exp\left(\frac{\mathcal{R}_B[A]}{\beta}\right) \approx 0$. So, $\ln\left(1 + \exp\left(\frac{\mathcal{R}_B[A]}{\beta}\right)\right) \approx \ln(1) = 0$. Therefore,

$$\hat{\mathcal{R}}_B[A] = \beta \cdot \text{softplus}\left(\frac{\mathcal{R}_B[A]}{\beta}\right) \approx \beta \cdot 0 = 0. \tag{15}$$

More precisely, as $\frac{\mathcal{R}_B[A]}{\beta} \to -\infty$, $\exp\left(\frac{\mathcal{R}_B[A]}{\beta}\right) \to 0^+$. Thus, $1 + \exp\left(\frac{\mathcal{R}_B[A]}{\beta}\right) \to 1^+$, and $\ln\left(1 + \exp\left(\frac{\mathcal{R}_B[A]}{\beta}\right)\right) \to 0^+$. Consequently, $\hat{\mathcal{R}}_B[A] \to 0^+$.

If $\mathcal{R}_B[A] = 0$ (at the boundary), then $\exp(0) = 1$, so

$$\hat{\mathcal{R}}_B[A] = \beta \ln(1 + \exp(0)) = \beta \ln(2). \tag{16}$$

As $\beta \to 0^+$ (which happens as $\mu^{(t)} \to 0^+$ during optimization), $\hat{\mathcal{R}}_B[A] \to 0^+$. This is consistent with the desired behavior where the barrier becomes increasingly sharp. The key is that for any fixed $\beta > 0$, if $\mathcal{R}_B[A]$ becomes sufficiently negative, $\hat{\mathcal{R}}_B[A]$ will approach 0 from the positive side.

This softplus-based surrogate $\hat{\mathcal{R}}_B[A]$ effectively smooths the hard constraint boundary, ensuring that the argument of the logarithm in the barrier term always remains positive. This allows SGD to navigate near the feasible region boundary more robustly, preventing numerical issues while still strongly penalizing constraint violations as $\mu^{(t)}$ (and thus $\beta$ if $\beta = \mu^{(t)}$) decreases. This modification maintains the theoretical consistency with the original interior-point method while enhancing the stability of practical optimization with stochastic gradients.

## B    TABULAR RESULTS FOR THE MAIN PAPER

The results presented in the main paper are obtained by preserving the original activations. We use a block size of 32 for the MXFP format. Sections C appendix provide more experiments with quantized activations (MXFP4) and an additional set of experiments using 6.25 as max bit constraint (comparable to MXFP6 effective bits). Due to the supernet's nature, models momentarily take up to a maximum of 2x their original size during the search phase before the final compression during the rounding stage of the algorithm.

### B.1    DETAILED EXPERIMENTAL SETUP

Table 1: Detailed Training Parameters. All experiments use Learning Rate = 0.1, Weight Decay = 0, Adam parameters ($\beta_1$=0.9, $\beta_2$=0.999, $\epsilon$=1e-8), linear LR scheduler, 10 epochs, dataset = allenai/c4, and 128 samples for "fewshots" experiments, 256 samples for "zeroshot" experiments. All of the original model weights and parameters are frozen.

| Model | Size | Max Bits | $\mu^{(0)}$ | $\delta$ | $A^{l^{(0)}}$ | w_dtypes | Runtime |
|---|---|---|---|---|---|---|---|
| opt-125m | 125M | 4.5 | 0.50 | 0.2 | [0.95, 0.05] | mxfp4, mxfp8 | 2.6m |
| opt-350m | 350M | 4.5 | 0.50 | 0.5 | [0.95, 0.05] | mxfp4, mxfp8 | 6.9m |
| Qwen2.5-0.5B | 500M | 4.5 | 0.50 | 0.2 | [0.95, 0.05] | mxfp4, mxfp8 | 9.3m |
| gemma-3-1b-it | 1.0B | 4.5 | 0.25 | 0.5 | [0.95, 0.05] | mxfp4, mxfp8 | 18.2m |
| Llama-3.2-1B | 1.0B | 4.5 | 0.25 | 0.5 | [0.95, 0.05] | mxfp4, mxfp8 | 20.5m |
| opt-1.3b | 1.3B | 4.5 | 0.25 | 0.5 | [0.95, 0.05] | mxfp4, mxfp8 | 22.1m |
| Qwen2.5-1.5B | 1.5B | 4.5 | 0.50 | 0.2 | [0.95, 0.05] | mxfp4, mxfp8 | 26.7m |
| gemma-2b | 2.0B | 4.5 | 0.25 | 0.5 | [0.95, 0.05] | mxfp4, mxfp8 | 39.6m |
| Llama-3.2-3B | 3.0B | 4.5 | 0.25 | 0.5 | [0.85, 0.15] | mxfp4, mxfp8 | 52.3m |
| Qwen2.5-3B | 3.0B | 4.5 | 0.25 | 0.5 | [0.95, 0.05] | mxfp4, mxfp8 | 52.8m |
| gemma-3-4b-it | 4.0B | 4.5 | 0.50 | 0.2 | [0.95, 0.05] | mxfp4, mxfp8 | 32.1m |

## B.2   DETAILED FEWSHOT RESULTS

Table 2: Results for **C4 Perplexity** (**lower is better**).  Color indicates our method outperforms: MXFP4 , MXFP6 , or MXFP8 baseline.

| Model | Size | MXFP4 | MXFP6 | MXFP8 | Mixed 4.5 bits |
|---|---|---|---|---|---|
| opt-125m | 125M | 29.25 | 26.16 | 25.98 | 28.98 |
| opt-350m | 350M | 24.79 | 22.20 | 21.96 | 22.05 |
| Qwen2.5-0.5B | 500M | 23.60 | 20.56 | 20.05 | 21.09 |
| gemma-3-1b-it | 1.0B | 42.06 | 37.24 | 35.78 | 35.48 |
| Llama-3.2-1B | 1.0B | 19.13 | 16.51 | 16.11 | 16.21 |
| opt-1.3b | 1.3B | 16.34 | 15.72 | 15.65 | 15.71 |
| Qwen2.5-1.5B | 1.5B | 15.58 | 14.25 | 14.10 | 15.95 |
| gemma-2b | 2.0B | 14.71 | 13.69 | 13.51 | 14.15 |
| Llama-3.2-3B | 3.0B | 15.28 | 13.97 | 13.68 | 12.66 |
| Qwen2.5-3B | 3.0B | 12.97 | 12.28 | 12.35 | 13.69 |
| gemma-3-4b-it | 4.0B | 25.86 | 22.80 | 22.97 | 18.59 |

Table 3: Results for **C4 Accuracy** (**higher is better**).  Color indicates our method outperforms: MXFP4 , MXFP6 , or MXFP8 baseline.

| Model | Size | MXFP4 | MXFP6 | MXFP8 | Mixed 4.5 bits |
|---|---|---|---|---|---|
| opt-125m | 125M | 37.9% | 39.3% | 39.4% | 36.0% |
| opt-350m | 350M | 40.0% | 41.4% | 41.5% | 41.5% |
| Qwen2.5-0.5B | 500M | 38.1% | 39.9% | 39.3% | 40.7% |
| gemma-3-1b-it | 1.0B | 36.1% | 37.8% | 38.1% | 36.5% |
| Llama-3.2-1B | 1.0B | 41.5% | 43.6% | 44.0% | 42.4% |
| opt-1.3b | 1.3B | 44.4% | 44.9% | 45.0% | 44.9% |
| Qwen2.5-1.5B | 1.5B | 43.0% | 43.6% | 43.7% | 43.9% |
| gemma-2b | 2.0B | 44.9% | 45.5% | 46.1% | 44.9% |
| Llama-3.2-3B | 3.0B | 44.6% | 45.9% | 46.2% | 46.0% |
| Qwen2.5-3B | 3.0B | 44.4% | 45.6% | 45.8% | 45.8% |
| gemma-3-4b-it | 4.0B | 42.1% | 43.2% | 43.4% | 44.5% |

## B.3   DETAILED ZEROSHOT RESULTS

Table 4: Results for **LAMBADA** (**higher is better**).  Color indicates our method outperforms: MXFP4 , MXFP6 , or MXFP8 baseline.

| Model | Size | MXFP4 | MXFP6 | MXFP8 | Mixed 4.5 bits |
|---|---|---|---|---|---|
| opt-125m | 125M | 37.6% | 37.5% | 37.1% | 37.8% |
| opt-350m | 350M | 40.2% | 45.8% | 44.8% | 44.9% |
| Qwen2.5-0.5B | 500M | 36.4% | 49.3% | 50.7% | 50.5% |
| gemma-3-1b-it | 1.0B | 38.5% | 43.1% | 42.9% | 45.0% |
| Llama-3.2-1B | 1.0B | 52.3% | 61.1% | 62.3% | 59.0% |
| opt-1.3b | 1.3B | 57.4% | 57.8% | 57.8% | 58.4% |
| Qwen2.5-1.5B | 1.5B | 59.7% | 63.2% | 62.0% | 60.4% |
| gemma-2b | 2.0B | 58.9% | 64.1% | 65.8% | 64.9% |
| Llama-3.2-3B | 3.0B | 68.0% | 70.5% | 70.4% | 69.7% |
| Qwen2.5-3B | 3.0B | 65.7% | 65.0% | 67.4% | 64.7% |
| gemma-3-4b-it | 4.0B | 56.4% | 59.5% | 59.6% | 57.7% |

Table 5: Results for **Wikitext Perplexity** (**lower is better**). Color indicates our method outperforms: MXFP4 , MXFP6 , or MXFP8 baseline.

| Model | Size | MXFP4 | MXFP6 | MXFP8 | Mixed 4.5 bits |
|-------|------|-------|-------|-------|----------------|
| opt-125m | 125M | 37.51 | 32.45 | 32.18 | 33.49 |
| opt-350m | 350M | 29.70 | 26.05 | 25.65 | 25.67 |
| Qwen2.5-0.5B | 500M | 22.07 | 18.29 | 17.83 | 18.00 |
| gemma-3-1b-it | 1.0B | 36.69 | 30.83 | 29.56 | 27.71 |
| Llama-3.2-1B | 1.0B | 14.25 | 11.89 | 11.67 | 12.53 |
| opt-1.3b | 1.3B | 17.42 | 16.55 | 16.46 | 16.53 |
| Qwen2.5-1.5B | 1.5B | 13.76 | 12.33 | 12.18 | 12.62 |
| gemma-2b | 2.0B | 45.47 | 94.84 | 42.10 | 42.57 |
| Llama-3.2-3B | 3.0B | 10.23 | 9.42 | 9.32 | 9.83 |
| Qwen2.5-3B | 3.0B | 15.97 | 15.03 | 12.77 | 15.28 |
| gemma-3-4b-it | 4.0B | 18.86 | 18.74 | 18.73 | 17.05 |

## C  MXFP4 ACTIVATIONS

### C.1  EXPERIMENTAL SETUP

Table 6: Detailed Training Parameters. All experiments use Learning Rate = 0.1, Weight Decay = 0, Adam parameters ($\beta_1$=0.9, $\beta_2$=0.999, $\epsilon$=1e-8), linear LR scheduler, 10 epochs, dataset = wikitext-2-raw, and 128 256 samples. All of the original model weights and parameters are frozen.

| Model | Size | Max Bits | $\mu^{(0)}$ | $\delta$ | $A^{l\,(0)}$ | w_dtypes | Runtime |
|-------|------|----------|-------------|----------|--------------|----------|---------|
| opt-125m | 125M | 4.5 | 0.50 | 0.5 | [0.95, 0.05] | mxfp4, mxfp8 | 3.3m |
| opt-125m | 125M | 6.25 | 0.50 | 0.5 | [0.8, 0.2] | mxfp4, mxfp8 | 3.3m |
| opt-350m | 350M | 4.5 | 0.50 | 0.5 | [0.95, 0.05] | mxfp4, mxfp8 | 8.7m |
| opt-350m | 350M | 6.25 | 0.50 | 0.5 | [0.8, 0.2] | mxfp4, mxfp8 | 8.7m |
| Qwen2.5-0.5B | 500M | 4.5 | 0.50 | 0.5 | [0.95, 0.05] | mxfp4, mxfp8 | 11.5m |
| Qwen2.5-0.5B | 500M | 6.25 | 0.50 | 0.5 | [0.8, 0.2] | mxfp4, mxfp8 | 11.5m |
| Llama-3.2-1B | 1.0B | 4.5 | 0.50 | 0.5 | [0.95, 0.05] | mxfp4, mxfp8 | 23.1m |
| Llama-3.2-1B | 1.0B | 6.25 | 0.50 | 0.5 | [0.8, 0.2] | mxfp4, mxfp8 | 23.1m |
| gemma-3-1b-pt | 1.0B | 4.5 | 0.50 | 0.5 | [0.95, 0.05] | mxfp4, mxfp8 | 21.3m |
| gemma-3-1b-pt | 1.0B | 6.25 | 0.50 | 0.5 | [0.8, 0.2] | mxfp4, mxfp8 | 21.3m |
| Qwen2.5-1.5B | 1.5B | 4.5 | 0.50 | 0.5 | [0.95, 0.05] | mxfp4, mxfp8 | 31.0m |
| Qwen2.5-1.5B | 1.5B | 6.25 | 0.50 | 0.5 | [0.8, 0.2] | mxfp4, mxfp8 | 31.0m |
| gemma-2-2b | 2.0B | 4.5 | 0.50 | 0.5 | [0.95, 0.05] | mxfp4, mxfp8 | 49.4m |
| gemma-2-2b | 2.0B | 6.25 | 0.50 | 0.5 | [0.8, 0.2] | mxfp4, mxfp8 | 49.5m |
| Llama-3.2-3B | 3.0B | 4.5 | 0.50 | 0.5 | [0.95, 0.05] | mxfp4, mxfp8 | 1.0h |
| Llama-3.2-3B | 3.0B | 6.25 | 0.50 | 0.5 | [0.8, 0.2] | mxfp4, mxfp8 | 1.0h |
| Qwen2.5-3B | 3.0B | 4.5 | 0.50 | 0.5 | [0.95, 0.05] | mxfp4, mxfp8 | 59.3m |
| Qwen2.5-3B | 3.0B | 6.25 | 0.50 | 0.5 | [0.8, 0.2] | mxfp4, mxfp8 | 59.3m |

## C.2 DETAILED FEWSHOTS RESULTS

Table 7: Results for **WikiText-2-raw Perplexity** (**lower is better**). Color indicates our method outperforms: MXFP4 , MXFP6 , or MXFP8 baseline.

| Model | Size | MXFP4 | MXFP6 | MXFP8 | Mixed 4.5 | Mixed 6.25 |
|-------|------|-------|-------|-------|-----------|------------|
| opt-125m | 125M | 109.55 | 80.91 | 79.62 | 90.56 | 38.41 |
| opt-350m | 350M | 68.39 | 55.07 | 53.74 | 58.96 | 28.95 |
| Qwen2.5-0.5B | 500M | 33.23 | 21.28 | 20.13 | 23.55 | 12.58 |
| Llama-3.2-1B | 1.0B | 50.35 | 29.09 | 27.28 | 32.12 | 20.52 |
| gemma-3-1b-pt | 1.0B | 29.91 | 25.75 | 25.28 | 27.69 | 18.31 |
| Qwen2.5-1.5B | 1.5B | 14.48 | 12.02 | 11.89 | 13.20 | 17.46 |
| gemma-2-2b | 2.0B | 38.60 | 35.72 | 33.38 | 33.70 | 14.19 |
| Llama-3.2-3B | 3.0B | 44.97 | 66.66 | 53.52 | 27.94 | 15.73 |
| Qwen2.5-3B | 3.0B | 11.44 | 10.27 | 10.04 | 10.92 | 14.38 |

Table 8: Results for **WikiText-2-raw Accuracy** (**higher is better**). Color indicates our method outperforms: MXFP4 , MXFP6 , or MXFP8 baseline.

| Model | Size | MXFP4 | MXFP6 | MXFP8 | Mixed 4.5 | Mixed 6.25 |
|-------|------|-------|-------|-------|-----------|------------|
| opt-125m | 125M | 25.2% | 27.9% | 28.1% | 26.9% | 32.8% |
| opt-350m | 350M | 28.9% | 32.0% | 32.2% | 30.4% | 36.0% |
| Qwen2.5-0.5B | 500M | 36.9% | 41.8% | 42.5% | 40.6% | 49.8% |
| Llama-3.2-1B | 1.0B | 34.7% | 40.5% | 41.4% | 39.2% | 40.5% |
| gemma-3-1b-pt | 1.0B | 40.1% | 42.1% | 42.1% | 40.9% | 41.9% |
| Qwen2.5-1.5B | 1.5B | 46.3% | 48.6% | 48.8% | 47.3% | 41.7% |
| gemma-2-2b | 2.0B | 39.1% | 40.5% | 41.1% | 40.4% | 44.9% |
| Llama-3.2-3B | 3.0B | 36.0% | 32.7% | 34.6% | 40.8% | 44.0% |
| Qwen2.5-3B | 3.0B | 49.1% | 50.6% | 51.0% | 49.7% | 42.3% |

## C.3 DETAILED ZEROSHOT RESULTS

Table 9: Results for **ARC-Challenge** (**higher is better**). Color indicates our method outperforms: MXFP4 , MXFP6 , or MXFP8 baseline.

| Model | Size | MXFP4 | MXFP6 | MXFP8 | Mixed 4.5 | Mixed 6.25 |
|---|---|---|---|---|---|---|
| opt-125m | 125M | 20.0% | 19.6% | 19.7% | 19.2% | 19.5% |
| opt-350m | 350M | 20.6% | 20.1% | 20.4% | 21.1% | 20.6% |
| Qwen2.5-0.5B | 500M | 27.0% | 27.6% | 28.9% | 28.4% | 28.5% |
| Llama-3.2-1B | 1.0B | 28.4% | 31.9% | 31.6% | 29.4% | 31.7% |
| gemma-3-1b-pt | 1.0B | 33.4% | 34.6% | 35.4% | 34.0% | 35.5% |
| Qwen2.5-1.5B | 1.5B | 40.4% | 40.5% | 42.2% | 39.9% | 41.2% |
| gemma-2-2b | 2.0B | 43.7% | 45.6% | 46.8% | 44.5% | 45.2% |
| Llama-3.2-3B | 3.0B | 41.3% | 42.2% | 41.7% | 40.5% | 42.1% |
| Qwen2.5-3B | 3.0B | 43.3% | 44.0% | 45.2% | 44.1% | 45.7% |

Table 10: Results for **LAMBADA** (**higher is better**). Color indicates our method outperforms: MXFP4 , MXFP6 , or MXFP8 baseline.

| Model | Size | MXFP4 | MXFP6 | MXFP8 | Mixed 4.5 | Mixed 6.25 |
|---|---|---|---|---|---|---|
| opt-125m | 125M | 37.6% | 37.5% | 37.1% | 37.3% | 36.2% |
| opt-350m | 350M | 40.2% | 45.8% | 44.8% | 42.0% | 43.3% |
| Qwen2.5-0.5B | 500M | 36.4% | 49.3% | 50.7% | 44.1% | 47.6% |
| Llama-3.2-1B | 1.0B | 52.3% | 61.1% | 62.3% | 54.1% | 62.2% |
| gemma-3-1b-pt | 1.0B | 53.0% | 56.1% | 55.8% | 53.9% | 56.8% |
| Qwen2.5-1.5B | 1.5B | 59.7% | 63.2% | 62.0% | 61.2% | 62.3% |
| gemma-2-2b | 2.0B | 64.9% | 69.5% | 70.0% | 66.5% | 69.0% |
| Llama-3.2-3B | 3.0B | 68.0% | 70.5% | 70.4% | 68.9% | 69.9% |
| Qwen2.5-3B | 3.0B | 65.7% | 65.0% | 67.4% | 68.4% | 66.8% |

# D ADDITIONAL EXPERIMENTS

To demonstrate the robustness of our method across varying hardware constraints, we evaluated the performance of our Constrained Mixed Precision Search (CMPS) against uniform baselines across a continuous range of effective bit-widths.

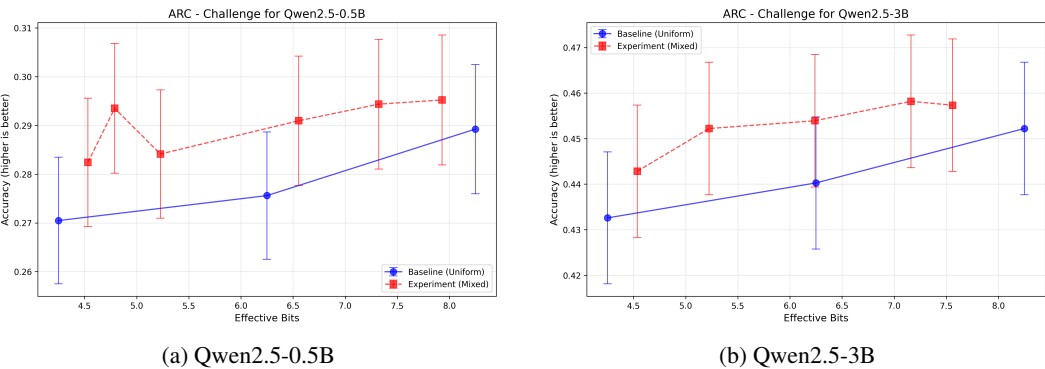

(a) Qwen2.5-0.5B                    (b) Qwen2.5-3B

Figure 6: **ARC-Challenge Accuracy vs. Effective Bits.** Comparison of our Mixed-Precision Search (Red) vs. Uniform Baselines (Blue). The mixed-precision approach consistently yields a superior Pareto frontier, achieving higher accuracy for the same effective bit budget across both model sizes.

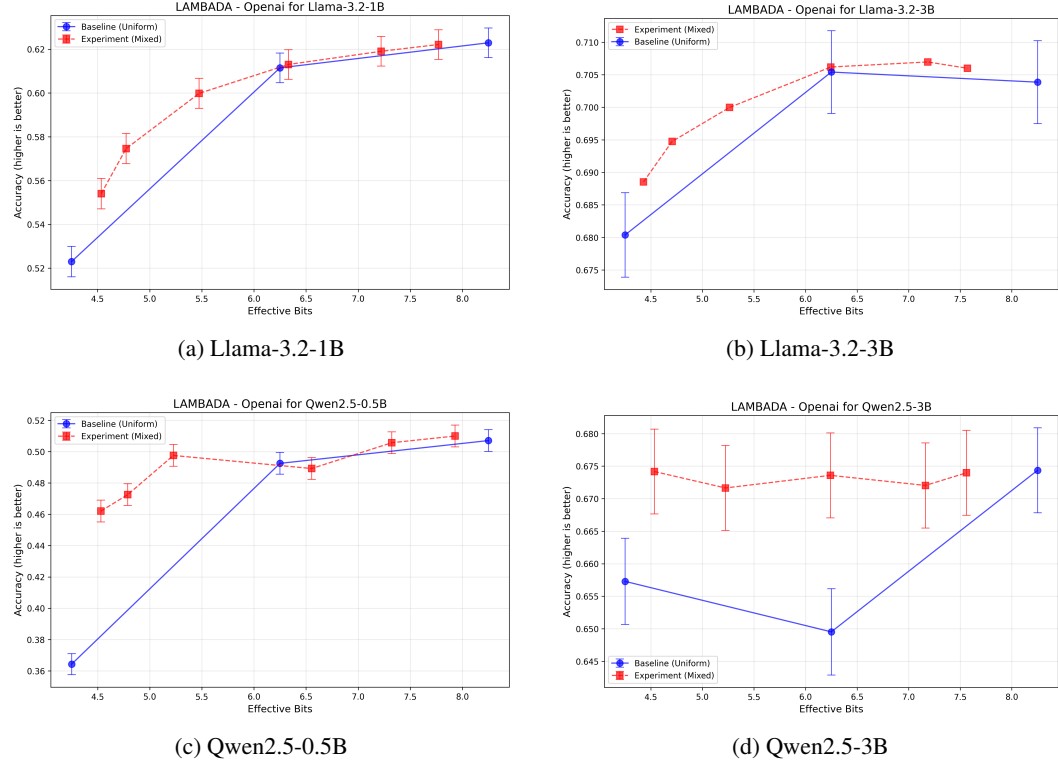

(a) Llama-3.2-1B                    (b) Llama-3.2-3B

(c) Qwen2.5-0.5B                    (d) Qwen2.5-3B

Figure 7: **LAMBADA (OpenAI) Accuracy vs. Effective Bits.** Our constrained optimization (Red dashed line) consistently outperforms the uniform quantization baseline (Blue solid line). Notably, our method frequently achieves the performance of higher-precision uniform models (e.g., 6-bit) while using significantly fewer bits (e.g., 4.5 bits).

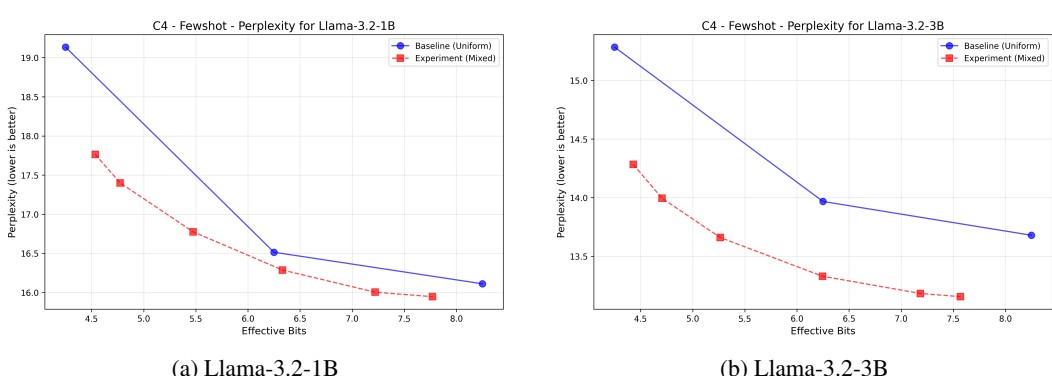

(a) Llama-3.2-1B  (b) Llama-3.2-3B

Figure 8: **C4 Validation Perplexity vs. Effective Bits (Lower is Better).** The mixed-precision allocation provides a strictly better trade-off curve, achieving lower perplexity at every measured bit-width constraint compared to uniform MXFP quantization.

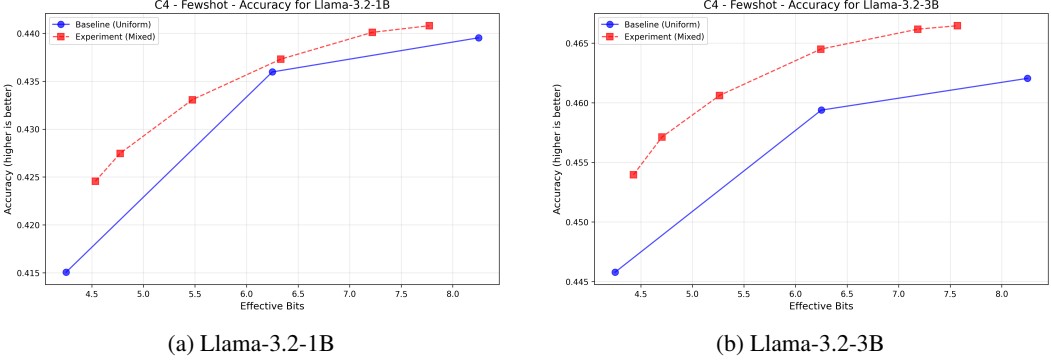

(a) Llama-3.2-1B  (b) Llama-3.2-3B

Figure 9: **C4 Validation Accuracy vs. Effective Bits.** Similar to the perplexity results, the accuracy metric on the calibration domain (C4) shows that the learned allocation preserves model capability better than uniform baselines as the compression rate increases.

