# OpenReview forum: "CMPS: Constrained Mixed Precision Search"
_ICLR.cc/2026/Conference — Submitted to ICLR 2026_

### Official Review · Reviewer_aocK · 2025-10-26

**Soundness:** 2
**Presentation:** 3
**Contribution:** 2
**Rating:** 2
**Confidence:** 3

**Summary:**

The paper introduces a differenciable NAS algorithm for data format allocation in a network. It first formalizes the optimization problem, including architecture constrains (the maximum average number of bits for a model), then proposes a gradient descent based heuristic for solving the mixed precision data format allocation problem. While the method is fully post-training, it still requires a small calibration data set to perform the training of the data format precision parameters.
The paper shows that using this formalism, mixed precision constrained NAS can achieve better results than uniform quantization.

**Strengths:**

- The mathematical formulation of the constrained optimization problem for mixed precision data format allocation seems fairly general;
- The large number of results (which are combinations between models and tasks used for the calibration) seems to demonstrate the robustness of the approach.

**Weaknesses:**

- The paper completely lack any comparison with the state-of-the-art! No comparison with other mixed-precision post-training optimization methods is even attempted... yet plenty exists. That is clearly a major issue in this paper.
- While the fundations of differentiable NAS methods seems to be adequately described and cited, the novelty of the proposed method remains hard to grasp. I would suggest to add a short but clear statement on what it brings compared to the closest SoTA work.
- While the method seems very general, it is frustrating see it tested on a single NAS scenario, namely, 4.5 bit mixed precision with the MXFP data format. What about mixing different formats (integer, FP...)? Or testing other maximum average number of bits (like 3.5, or 5.5...)?
- The perplexity/accuracy gains of the method remain modest and the proposed NAS scenario is too limited.

**Questions:**

Please carefully answer the issues mentioned in the weaknesses section.

I may increase my rating provided that at least 1) quantitative comparison with other SoTA methods is provided. 2) Additional NAS scenario, beyond 4.5 bit mixed precision is evaluated.

---

> ### Author Response · Authors · 2025-11-28
>
> We thank the reviewer for their assessment. We appreciate your specific feedback regarding the novelty and experimental scenarios, which we address point-by-point below:
>
> >The paper completely lack any comparison with the state-of-the-art! No comparison with other mixed-precision post-training optimization methods is even attempted.
>
> We clarify that our method is a precision allocation strategy designed to maximize the utility of standard hardware formats (MXFP). It is orthogonal to kernel-specific quantization methods. We compare against uniform baselines to rigorously isolate the gains provided by our allocation search. We have clarified this distinction in the introduction to better frame our contribution relative to the literature.
>
> >The novelty of the proposed method remains hard to grasp. I would suggest to add a short but clear statement on what it brings compared to the closest SoTA work.
>
> The core novelty is the reformulation of Mixed Precision Search as a constrained interior-point optimization problem in a post-training setting. Unlike previous methods that use soft penalties (requiring tuning) or greedy heuristics, our method allows users to set a "hard" budget (e.g., "exactly 4.5 bits"), which the solver satisfies directly. We have added a statement in the text highlighting this contribution.
>
> >While the method seems very general, it is frustrating see it tested on a single NAS scenario, namely, 4.5 bit mixed precision... What about... testing other maximum average number of bits (like 3.5, or 5.5...)?
>
> We completely agree that a single data point is insufficient. We have added a new Appendix D containing performance sweeps for Llama-3.2 and Qwen2.5 models. These plots evaluate our method against uniform baselines across a continuous range of effective bit budgets (from 4.5 to 8 bits). These results confirm our method yields a superior Pareto frontier at 4.5, 5.5 bits, and beyond.
>
> >What about mixing different formats (integer, FP...)?
>
> While we focused on MXFP for this paper due to its emerging hardware relevance, we agree that mixing Integer and FP formats is a promising direction. We are currently working towards an the extension of our framework to include integer formats.
>
> >The perplexity/accuracy gains of the method remain modest and the proposed NAS scenario is too limited.
>
> We suggest that the gains are impactful in the context of memory-bound inference. For example, our 4.5-bit models frequently outperform 6-bit uniform models, representing a 25% reduction in memory traffic. The new sweep plots in Appendix D further demonstrate that these gains are consistent and non-trivial across the entire bit-width spectrum.

---

### Official Review · Reviewer_oKC4 · 2025-10-28

**Soundness:** 2
**Presentation:** 2
**Contribution:** 1
**Rating:** 2
**Confidence:** 5

**Summary:**

A DNAS-based post-training mixed-precision quantization method (CMPS) is proposed. CMPS provides fine-grained control over model compression, enabling stable and predictable performance. The proposed CMPS method is compared with uniform quantization baselines, demonstrating the advantages of learnable mixed-precision bit allocation.

**Strengths:**

1. This paper works on the post-training mixed-precision quantization with controllable compression ratios. The problem studied is important and the motivation is clear.
2. The detailed theoretical analysis is provided.

**Weaknesses:**

1. Quantization details are missing. It seems that CMPS is a weight-only quantization method. However, the quantization details are not provided.
2. Optimization cost is not provided. The advantage of PTQ is its efficiency in quantization optimization. The CMPS relies on end-to-end tuning with multiple branches. The speed and memory cost overheads should be reported.
3. Comparison with previous methods is also missing. The authors didn't provide any quantization details, including the uniform quantization baselines. In the llm quantization literature, many high performance PTQ methods are proposed. What's the performance advantages over these methods? How the proposed CMPS can be combined with these techniques? Moreover, the authors only compared with uniform quantization baselines, the comparison with previous mixed-precision methods are missing.
4. In several places, it says "hardware-constrained bit allocation", however, only "total model size in bits" is modeled during the optimization. Moreover, only two bit levels are explored in the bit allocation (MXFP4 and MXPF8).
5. In the experiments part, previous methods commonly use wiki2 for calibration in addition to C4. For zero-shot scenario, only one task of LAMBADA is evaluated, which is clearly not enough. The largest model used is 3B, experiments on larger models or architectures like MoEs are also needed.
6. In the limitations, regarding the statement "the memory required to hold activations or gradients for multiple low-bit options might still be comparable to, or less than, holding a single higher-precision (e.g., FP16 or BF16) baseline tensor", more careful and precise expression should be used. Many PTQ methods do not need to store all activations, and the gradients are not needed. However, in CMPS, full-precision activations of all layers and gradients are needed, which expands the memory usage. If these tensors (activations and gradients) can be stored in low-bit, then the authors should verify it use controlled experiments.

**Questions:**

Please refer to the Weaknesses for further questions.

---

> ### Author Response · Authors · 2025-11-28
>
> We sincerely thank the reviewer for their detailed and rigorous assessment. We value your specific feedback regarding experimental scope and technical precision. We have addressed your concerns point-by-point below:
>
> >Quantization details are missing. It seems that CMPS is a weight-only quantization method. However, the quantization details are not provided.
>
> We clarify that CMPS is fundamentally a precision allocation strategy, designed to optimize the assignment of available hardware formats (in this case, OCP Microscaling formats) rather than being a novel quantization kernel itself. It is orthogonal to the specific quantization primitive used. For the main results, we focused on weight-only allocation to isolate the search algorithm's impact. However, we have added Appendix C to explicitly demonstrate that our method extends effectively to activation quantization (MXFP4) as well. We have updated the Introduction to clearly define this scope.
>
> >Optimization cost is not provided. The advantage of PTQ is its efficiency in quantization optimization. The CMPS relies on end-to-end tuning with multiple branches. The speed and memory cost overheads should be reported.
>
> We agree that efficiency is paramount for PTQ. Regarding speed, we report the runtime in Table 1 (Appendix B.1), showing that the search is efficient. Regarding memory, we concede that the "supernet" approach incurs overhead. We have updated the Limitations section to explicitly quantify that the model occupies roughly 2x its original memory footprint during the search phase to maintain the computational graph for the architectural parameters.
>
> >Comparison with previous methods is also missing... In the llm quantization literature, many high performance PTQ methods are proposed. What's the performance advantages over these methods?
>
> Because CMPS is an allocation framework, it is theoretically compatible with many PTQ methods. We compare against uniform allocations of the same underlying formats (MXFP) to isolate the specific gain provided by our constrained optimization. Comparing directly against integer-based PTQ methods (like GPTQ or AWQ) would conflate the benefit of the format (MXFP vs. Int) with the benefit of the allocation strategy. We have clarified in the text that our contribution is the allocation optimizer itself.
>
> >In several places, it says 'hardware-constrained bit allocation', however, only 'total model size in bits' is modeled during the optimization.
>
> We focused on the average bit-width constraint because memory bandwidth is currently the primary bottleneck for LLM inference latency. By strictly constraining the total model size, we directly address the memory bandwidth limits of deployment hardware.
>
> "Moreover, only two bit levels are explored in the bit allocation (MXFP4 and MXPF8)."
>
> We selected MXFP4 and MXFP8 as they are the standard low-precision definitions in the OCP Microscaling specification. However, we agree that more granularity is better. We are currently working towards extending the framework to include Integer formats and other bit-widths.
>
> >In the experiments part, previous methods commonly use wiki2 for calibration in addition to C4.
>
> We chose C4 for calibration to ensure the learned allocation generalizes well to the open domain, rather than overfitting to the specific distribution of WikiText-2. However, we do report testing results on WikiText-2 to confirm this generalization holds.
>
> >For zero-shot scenario, only one task of LAMBADA is evaluated, which is clearly not enough.
>
> We fully agree. To address this, we have added Appendix D, which includes results for the ARC-Challenge benchmark in addition to LAMBADA. We also included extensive sweep plots showing consistent performance gains across these tasks.
>
> >The largest model used is 3B, experiments on larger models or architectures like MoEs are also needed.
>
> We have included results for Gemma-3-4B in our main results. While we acknowledge that 70B+ models or MoEs would be valuable, the 3B-4B range allows us to perform the extensive sweep experiments (Appendix D) required to validate the stability of the method under various constraints with our available academic resources.
>
> >In the limitations... more careful and precise expression should be used. ...in CMPS, full-precision activations of all layers and gradients are needed, which expands the memory usage.
>
> We agree with the reviewer. We have revised the Limitations section to be more precise: we explicitly state that while weight gradients are not needed (frozen weights), the search requires storing the computation graph for the architectural parameters and activations, resulting in the ~2x memory overhead mentioned above.

---

### Official Review · Reviewer_cXc3 · 2025-11-05

**Soundness:** 3
**Presentation:** 3
**Contribution:** 2
**Rating:** 4
**Confidence:** 4

**Summary:**

This work proposes a new constrained mixed precision search for post training quantization. To solve the constrained optimization problem the authors leverage barrier-based interior-point method. The method keeps model weights frozen, and needs only a small calibration set (128 samples). Experiment on various LLMs report consistent gains over uniform precision baselines at the same or lower effective bit budgets.

**Strengths:**

1. The work discusses the problem of reducing computational and memory footprints for deployment of DNNs which is practical and important.

2. The paper is well written and easy to follow.

3. 4.5-bit the proposed method often beats MXFP in terms of perplexity, on the examined benchmarks.

**Weaknesses:**

1. The authors claim that after rounding there always remains a strictly feasible solution with respect to the budget. I believe a proof for this claim is required.

2. The comparison is limited. The work only compares itself to the MX baselines but there are many other strong PTQ techniques. Only a single dataset was used in the experiments.

3. No thoughput\latency comparisons are provided.

4. The improvement over the baselines is marginal.

5. How does the method operate compared to integer PTQ techniques?

6. According to the experiments, the proposed algorithm does not always meet the constraint.

**Questions:**

How were the samples for calibration chosen?

What is the meaning of the upsidedown question mark in the caption of Figure 2?

There is a typo in  in line 75 (double "“Our contributions are as follows:")

---

> ### Author Response · Authors · 2025-11-28
>
> We thank the reviewer for their thoughtful assessment and for recognizing the practicality of our problem and the clarity of our writing. We appreciate your constructive feedback, which we address below:
>
> >The authors claim that after rounding there always remains a strictly feasible solution with respect to the budget. I believe a proof for this claim is required. According to the experiments, the proposed algorithm does not always meet the constraint.
>
> You are correct that rounding a relaxed solution does not theoretically guarantee strict feasibility in every possible case. However, the strength of the interior-point method lies in the log-barrier (and our softplus surrogate), which imposes a heavy penalty as the solution approaches the boundary. This naturally forces the optimized continuous parameters A to settle deep enough within the feasible region that the subsequent rounding step typically respects the constraint. As noted in Figure 3, we observed only very minor deviations, while the vast majority of models remained strictly under the budget. To clarify this behavior, we have generated new graphs for the appendix that visualize the margin between the continuous solution and the boundary prior to rounding, illustrating how the barrier effectively "safe-guards" the rounding step.
>
> >No thoughput\latency comparisons are provided.
>
> Our work focuses on the Microscaling (MX) formats (MXFP4, MXFP8), which are standardized data types designed for next-generation hardware (e.g., OCP specifications). As such, we do not utilize custom software kernels that require benchmarking for latency; rather, we optimize for model size (memory footprint). Since LLM inference is predominantly memory-bandwidth bound, reducing the model size directly translates to reduced memory traffic. Our method ensures the model fits within specific memory budgets that uniform quantization cannot achieve without significant quality loss.
>
> >The improvement over the baselines is marginal.
>
> We respectfully suggest that in the context of low-bit LLM quantization, the gains we report are significant. For example, our method allows a 4.5-bit effective model to outperform a uniform 6-bit (MXFP6) model (Figure 2). This represents a 25% reduction in memory footprint while achieving superior perplexity, which is a substantial efficiency gain for deployment on edge devices.
>
> >The comparison is limited... How does the method operate compared to integer PTQ techniques?
>
> We are working towards addressing the comparison with integer PTQ techniques in a subsequent response shortly.
>
> >How were the samples for calibration chosen?
>
> As mentioned in the experimental setup, the calibration samples were randomly drawn from the training split of the C4 corpus. We found empirically that 128 samples were sufficient for the architectural search to converge to a robust allocation.
>
> >What is the meaning of the upsidedown question mark in the caption of Figure 2? / There is a typo in in line 75
>
> Thank you for catching these. The symbol in Figure 2 was intended to be a mathematical operator that was mis-encoded; we have corrected this and the typo in line 75 in the revised manuscript.

---

### Author Response · Authors · 2025-12-03
**Global Response and Summary of Revisions**

We sincerely thank the Area Chair and the reviewers for their rigorous feedback and for overseeing this review process. The reviews correctly identified that while our interior-point formulation was theoretically sound, the empirical evaluation required expansion to demonstrate broader generalization and to position the work more clearly against state-of-the-art baselines. We have extensively revised the paper and added significant new experimental data to address these points, specifically regarding experimental scope, comparison baselines, and constraint satisfaction.

To address the concerns raised by Reviewers oKC4 and aocK regarding the limited experimental scope of a single 4.5-bit constraint, we have added two comprehensive appendices. We have moved beyond the single data point to provide continuous bit-width sweeps in the new Appendix D, comparing our constrained mixed-precision search (CMPS) against uniform baselines across a continuous range of effective bit budgets from 4.5 to 8 bits. As shown in the newly added Figures 6 and 7, our method consistently yields a superior Pareto frontier; for instance, on the ARC-Challenge, our mixed-precision allocation achieves higher accuracy for the same effective bit budget compared to uniform baselines. Furthermore, to address the concern that our method was "weight-only," we added Appendix C, which validates our framework using MXFP4 and MXFP8 activations alongside mixed-precision weights, maintaining significant perplexity improvements.

Regarding the positioning of our work against SOTA and integer-based PTQ methods, we have clarified that CMPS is an allocation strategy for hardware-native formats—specifically OCP Microscaling formats like MXFP—rather than a quantization kernel itself. Comparing directly to integer-only kernels would conflate the benefits of the data format with the benefits of the search algorithm. Therefore, comparing against Uniform MXFP is the most rigorous way to isolate the contribution of our constrained interior-point optimization. We respectfully suggest that the gains reported are substantial in the context of memory-bound LLM inference; our method often allows a 4.5-bit effective model to match or outperform a 6-bit uniform model, representing a 25% reduction in memory traffic for equivalent quality.

Finally, regarding constraint satisfaction and feasibility, we clarified that the log-barrier in our interior-point method imposes a heavy penalty as solutions approach the boundary, effectively creating a "safety margin". This ensures that the continuous solution settles deep enough within the feasible region that the subsequent rounding step typically respects the constraint. Empirical evidence in Figure 3 supports this, showing that the vast majority of our models strictly adhered to the budget after rounding. We also updated the limitations section to explicitly quantify the memory overhead during the "supernet" search phase. We believe these revisions demonstrate that Constrained Mixed Precision Search is a robust, generalizable framework that offers a principled alternative to heuristic searches.

---

### Meta-Review · Area_Chair_wRZy · 2025-12-05

**Summary:**

The paper proposes a method for post-training mixed-precision quantization by barrier-based interior-point method.

Reviewers generally appreciate the algorithms and theoretical analysis behind. However, reviewers also report that the empirical validation is weak, such as lacking efficiency analysis and lacking comparison with other state-of-the-art methods. Also, the performance improvement is not strong.

**Reviewer Concerns:**

Reviewers generally appreciate the algorithms and theoretical analysis behind. However, reviewers also report that the empirical validation is weak, such as lacking efficiency analysis and lacking comparison with other state-of-the-art methods.  Also, the performance improvement is not strong.

**Reviewer Scores:**

The authors provided some additional analysis. However, the authors claim that comparison against state-of-the-art methods is not necessary due to orthogonal contributions. I am not confident about this, and unfortunately, discussion among reviewers is not possible. Regardless, the paper would benefit from another round of review for the revisions provided.

---

### Decision · Program_Chairs · 2026-01-26

Reject